# A pH Dual-Responsive Multifunctional Nanoparticle Based on Mesoporous Silica with Metal-Polymethacrylic Acid Gatekeeper for Improving Plant Protection and Nutrition

**DOI:** 10.3390/nano12040687

**Published:** 2022-02-18

**Authors:** Hua Pan, Weilan Huang, Litao Wu, Qihao Hong, Zhongxuan Hu, Meijing Wang, Fang Zhang

**Affiliations:** Faculty of Environment and Life, Beijing University of Technology, Beijing 100124, China; panh@emails.bjut.edu.cn (H.P.); huangweilan1226@163.com (W.H.); wltuiao@163.com (L.W.); hqh960617@163.com (Q.H.); slytherinkuma@foxmail.com (Z.H.); meijingwang131@163.com (M.W.)

**Keywords:** prochloraz, mesoporous silica nanoparticles, pH-responsive release, micronutrient, biosafety

## Abstract

Integrating pesticides and mineral elements into a multi-functional stimuli-responsive nanocarrier can have a synergistic effect on protecting plants from pesticides and the supply of nutrients. Herein, a pH dual-responsive multifunctional nanosystem regulated by coordination bonding using bimodal mesoporous silica (BMMs) as a carrier and coordination complexes of ferric ion and polymethacrylic acid (PMAA/Fe^3+^) as the gatekeeper was constructed to deliver prochloraz (Pro) for the smart treatment of wilt disease (Pro@BMMs−PMAA/Fe^3+^). The loading capacity of Pro@BMMs−PMAA/Fe^3+^ nanoparticles (Nps) was 24.0% and the “PMMA/Fe^3+^” complexes deposited on the BMMs surface could effectively protect Pro against photodegradation. The nanoparticles possessed an excellent pH dual-responsive release behavior and better inhibition efficacy against *Rhizoctonia solani*. Fluorescence tracking experiments showed that Nps could be taken up and transported in fungi and plants, implying that non-systemic pesticides could be successfully delivered into target organisms. Furthermore, BMMS−PMAA/Fe^3+^ nanocarriers could effectively promote the growth of crop seedlings and had no obvious toxicological influence on the cell viability and the growth of bacteria. This study provides a novel strategy for enhancing plant protection against diseases and reducing the risk to the environment.

## 1. Introduction

The extensive use of pesticides to eliminate pests and improve crop yield is an inevitable choice to meet the growing demand for food [1]. However, the effective utilization rate of traditional pesticides on target organisms is less than 0.1% [2,3]. In addition, multi-frequency and large-area spraying of pesticides increases the cost of agriculture application, and also harms the environment and non-target organisms [4,5,6]. The development of nanotechnology provides new strategies for resolving these problems. In recent years, nanotechnology has seen growing potential application in some fields, such as agricultural production, plant protection, and plant nutrition, which includes the transportation and release of agrochemical and genetic material, and the prevention and treatment of fungal diseases and pests [7,8,9,10]. Developing controlled-release formulations (CRFs) of pesticides using nanotechnology is an effective way to promote plant growth and protection [2,11]. Many carriers (e.g., polymers [12,13], inorganic porous nanomaterials [14,15], layered double hydroxides [16], and metal-organic frameworks [17,18]) have been widely utilized to prepare nano-pesticides. However, the research is mainly focused on controlled-release performance and the stability and bioactivity of nanoparticles (Nps). Owing to the existence of biological barriers, such as epidermis, plant cuticles, sieve plates, and cell walls, nanocarriers that can be taken up and transported by plants are limited to relatively small nanomaterials (<100 nm) [19,20]. The uptake and transport behaviors of nanocarriers in plants have been, to date, poorly investigated. 

Bimodal mesoporous silicas (BMMs) comprise a new kind of mesoporous silica material with a double-channel structure, with worm-like holes of approximately 3 nm in size and spherical particle accumulation holes of approximately 10–30 nm in size, which is obviously different from traditional mesoporous materials [21,22]. It is possible for BMMs to be taken up and transported by plants due to the 20–50 nm size of their Nps. Realizing the effective uptake and transfer of nano-pesticides by plants through surface spraying or soil application is a potential solution to improve the efficiency of pesticide application [23,24,25,26]. Benefiting from their modifiable surface, tunable particle/pore size and large specific surface area, and excellent environmental compatibility, BMMs are very conducive to the adsorption and desorption of guest molecules, such as pesticides and fertilizer, which were utilized in pesticide delivery systems in our previous study [27,28].

As is generally known, inorganic metal ions (Fe^3+^, Mg^2+^, and Zn^2+^) play an important role in the growth of plants, and they have been widely used as plant nutritional agents in agriculture [17,29,30,31]. The multifunctional metal cross-linked nanocomplex was fabricated through a coordination–bond interaction between inorganic metal ions and carriers, delivering nutrient substances to favor the growth of plants [31,32,33]. Polymethacrylic acid (PMAA), which is a functional polymer rich in carboxyl groups (−COOH), has been extensively used as a gatekeeper for pH response. Unfortunately, using PMAA as a gated material presents the problem of premature release of guest molecules and a single function [34]. Owing to the existence of metal coordination between −COO^−^ and Fe^3+^, Fe^3+^ can be employed as the cross-linking agent of PMAA to form “PMAA/Fe^3+^” coatings, which possess excellent structural stability to limit the premature leakage of pesticides and changeable structural transformation characteristics in response to different pH levels of the external environment for realizing the smart release of pesticides [30,34,35]. In acid or alkaline conditions, H^+^ and OH^−^ can disrupt the “PMAA/Fe^3+^” coating to achieve smart release of pesticides. The pH value of the crop itself and the growing environment in the field may be more diverse and complex. For example, pH values of cucumber leaves, roots, and fruits were found to be 8.06, 6.98, and 6.13, respectively [36]. The soil in northwestern China is mostly alkaline, while the soil in the South is mostly acidic [37]. In particular, plant pathogens can secrete acids to help themselves colonize [38,39].

Prochloraz (Pro), a broad-spectrum, high-efficiency imidazole fungicide, has a significant antibacterial effect on ascomycetes and deuteromycetes of cereals and other crops [40]. However, the utilization of Pro to prevent foliar disease is limited due to poor conductivity and light instability [36,41]. In the work reported herein, an acid/base dual-responsive nanosystem using BMMs−PMAA/Fe^3+^ as carriers was constructed to deliver Pro for the prevention and cure of wilt disease (designated as Pro@BMMs−PMAA/Fe^3+^). The procedure is shown in Figure 1. The characteristics parameters, release behaviors, and bioactivity against *Rhizoctonia solani* were investigated. Translocation in the plant and fungi was also observed. Furthermore, the nutritional function of BMMs−PMAA/Fe^3+^ as an iron fertilizer for enhancing rice growth was demonstrated. This work explores the feasibility of nanocarriers as a platform for enhancing plant protection and environmental safety.

## 2. Materials and Methods

### 2.1. Materials

Methacrylic acid (MAA, AR) was purchased from Tianjin Fuchen Chemical Reagent Co., Ltd. (Tianjin, China). 5-Aminofluorescein (5-AF, 96%), 3-(trimethoxysilyl) propyl methacrylate (MPTES, 98%), and potassium persulfate (KPS, 99%) were obtained from the Aladdin Reagent Co., Ltd. (Shanghai, China). Cetyl trimethyl ammonium bromide (CTAB, 98%), tetraethyl orthosilicate (TEOS, 99.9%), ferric chloride (FeCl_3_, 99%), potassium bromide (KBr), and other organic solvents were purchased from Sinopharm Chemical Reagent Beijing Co., Ltd. (Beijing, China). Dialysis membrane (molecular weight cutoff: 3500 Da) was purchased from Beijing Kebiquan Biotechnology Co., Ltd. (Beijing, China). Prochloraz technical (Pro TC, 97.5%) was obtained from Beijing Mindleader Agroscience Co., Ltd. (Beijing, China). *Rhizoctonia solani* and rice seeds were kindly provided by the Institute of Chinese Academy of Agricultural Sciences. Deionized water (18 MΩ·cm^−1^) was prepared using a Milli-Q water purification system (Millipore, Milford, MA, USA).

### 2.2. Synthesis of Pro@BMMs−PMAA/Fe^3+^ Nps

#### 2.2.1. Synthesis of BMMs

CTAB (522.4 mg) was dispersed in 20.8 mL of distilled water, stirred at 40 °C for 30 min, and then TEOS (1.6 mL) was slowly added to the mixture. Subsequently, 0.48 mL of ammonia water was rapidly added and stirring was continued until the solution became a white gel. The resultant white gel was collected by vacuum filtration, washed several times with water, and dried at 105 °C overnight in an oven. The synthesized white powder was heated at 550 °C for 5 h to remove CTAB and obtain BMMs.

#### 2.2.2. Synthesis of Poly-Methacrylic Acid-Coated BMMs (BMMs−PMAA)

BMMs−poly-MAA (BMMs−PMAA) was synthesized by seeded precipitation polymerization. Firstly, 500 mg of BMMs was suspended in 50 mL of methylbenzene followed by the addition of 250 µL of MPTES with stirring for 4 h at 70 °C. Then, the product was washed several times by methanol to remove free MPTES and dried at 60 °C for 12 h to yield the intermediates product (BMMs−MPTES). Subsequently, 200 mg of BMMs−MPTES and 0.8 mL of MAA were suspended in 40 mL of aqueous solution with stirring at 70 °C for 1 h after degassing and filling with nitrogen. Finally, 40 mg of KPS was added and stirred continually at 70 °C for 12 h under a nitrogen atmosphere, and the product (BMMs−PMAA) was collected after drying at 60 °C for 12 h.

#### 2.2.3. Synthesis of Pesticide-Loaded Iron-Chelated Nps (Pro@BMMs−PMAA/Fe^3+^)

BMMs−PMAA (250 mg) was added to 50 mL of an aqueous Fe^3+^ solution (1.0 mg·mL^−1^) under mechanical agitation for 24 h. The BMMs−PMAA/Fe^3+^ was then separated from the above mixtures utilizing centrifugation. Subsequently, the crude products were washed with abundant water to remove excess Fe^3+^ and the final BMMs−PMAA/Fe^3+^ Nps were dried at 60 °C for 12 h. For Pro loading, 200 mg of BMMs−PMAA/Fe^3+^ was added to a hexane solution (40 mL) with a Pro concentration of 10 mg mL^−1^. The resultant mixture was kept at room temperature for 24 h. Pro-loaded BMMs−PMAA/Fe^3+^, denoted Pro@BMMs−PMAA/Fe^3+^, was collected by centrifugation and dried at 60 °C under vacuum for 12 h.

### 2.3. Preparation of 5-AF-Functionalized Nps (BMMs−PMAA/5-AF)

To track the uptake and translocation of BMMs−PMAA Nps in rice plants and fungi, BMMs−PMAA Nps were fluorescently labeled with 5-AF. Briefly, 200 mg of BMMs−PMAA was dispersed in 20 mL of deionized water, and then 40 mg of 1-(3-Dimethylaminopropyl)-3-ethylcarbodiimide hydrochloride (EDC) and 30 mg of N-hydroxysuccinimide (NHS) were added to activate the carboxyl groups of the Nps. The mixture was stirred at 0 °C for 30 min, and 50 mg of 5-AF was introduced into the above mixture. The reaction system was stirred for 12 h at room temperature in the dark, and the BMMs−PMAA/5-AF Nps were centrifuged at 6000 rpm for 10 min and washed with ethanol several times. Subsequently, the resultant Nps were dried at 60 °C overnight, and they were later stored at room temperature in the dark.

### 2.4. Characterizations of Pro@BMMs−PMAA/Fe^3+^ Nps

Scanning electron microscopy (SEM) micrographs and electron-dispersive spectroscopy (EDS) maps were acquired by a Tecnai G2, F20 S-TWIN transmission electron microscope (Zeiss, Oberkochen, Germany) operated at an accelerating voltage of 10.00 kV. N2 adsorption–desorption isotherms were carried out on an Autosorb-iQ pore analyzer (Quantachrome, Boynton Beach, FL, USA) at 77.3 K under continuous adsorption conditions. The specific surface areas were calculated by the Brunauer–Emmett–Teller (BET) model, and the pore size distribution was estimated from the adsorption data by the Barrett–Joyner–Halenda (BJH) model. Elements were analyzed using X-ray photoelectron spectroscopy (XPS) (ESCALAB 250Xi, Thermo Fisher Scientific, Waltham, MA, USA). X-ray powder-diffraction (XRPD) analysis was performed on an X-ray powder diffractometer (D8 ADVANCE X, Bruker/AXS, Karlsruhe, Germany) with Ni-filtered Cu Kα radiation. The chemical functional groups of samples were analyzed over the spectral region between 4000 and 400 cm^−1^ by a Nexus 470 Fourier-transform infrared spectroscopy (FTIR) spectrometer (Nicolet Nexus 470, Nicolet Instrument Corp., Concord, CA, USA) with a pellet of powdered potassium bromide. The zeta potential and sizes of samples were measured based on dynamic light scattering (DLS) by a ZetaSizer Nano ZS analyzer (Malvern Instruments Ltd., Malvern, UK). Thermogravimetric analysis (TGA) was carried out using a thermal analyzer (PerkinElmer, Waltham, MA, USA) under nitrogen atmosphere from 30 to 800 °C at a heating rate of 10 °C·min^−1^.

### 2.5. Pro Loading Content

To determine the loading content of Pro, 10 mg of Pro@BMMs−PMAA/Fe^3+^ was dispersed in 30 mL of methanol using ultrasonication for 3 h. After centrifugation at 7000 rpm for 10 min and filtration with a 0.22 μm filter membrane, the Pro concentration in the supernatant was measured using high-performance liquid chromatography (HPLC) with a diode array detector (1200-DAD, Agilent Corp., Santa Clara, CA, USA). The operating parameters for HPLC determination were as follows: C18 reversed-phase column (5 μm × 4.6 mm × 250 mm), column temperature of 30 °C, detection wavelength of 220 nm, mobile phase: acetonitrile/0.1% acetic acid (*v*/*v*, 70:30), injection volume of 10 μL, and a flow rate of 1.0 mL·min^−1^. The loading content (%) of Pro in Nps was calculated using the following equation:(1)$$\mathrm{loading}\;\mathrm{content}\;\left(\%\right)=\frac{{\mathrm{weight}\;\mathrm{of}\;\mathrm{Pro}\;\mathrm{in}\;\mathrm{Pro}@\mathrm{BMMs}-\mathrm{PMAA}/\mathrm{Fe}}^{3+}}{\mathrm{weight}\;\mathrm{of}\;\mathrm{Pro}@\mathrm{BMMs}-\mathrm{PMAA}/{\mathrm{Fe}}^{3+}}\times 100\%$$

### 2.6. In Vitro Release Behavior of Pro

To study the pesticide release behavior of Pro@BMMs−PMAA/Fe^3+^, the release media with different pH values (5.0, 7.0, and 9.0) containing 0.1% Tween-80 emulsifier were prepared separately. Approximately 10 mg of Pro@BMMs−PMAA/Fe^3+^ was dispersed in 2.0 mL of release medium in dialysis bags (molecular weight cutoff: 3500 Da). The sealed dialysis bags were thereafter placed in 48 mL of release medium at room temperature and stirred at 100 rpm using a vibrating table (Taizhou Nomi Medical Technology Co., Ltd., Taizhou, China). At designated time intervals, 1 mL of the solution was withdrawn for HPLC analysis, and an equal volume of fresh release medium was added to ensure constant volume. The experiment was carried out three times. The cumulative release (Cr) of Pro from the Pro@BMMs−PMAA/Fe^3+^ was calculated according to the following formulation:(2)$$C_{r}=\frac{V\sum_{i=0}^{n-1}C_{i}+V_{0}C_{n}}{m}$$
where *m* is the total quantity of pesticide loaded into Pro@BMMs−PMAA/Fe^3+^ (mg), V_0_ is the volume of the sustained-release solution, and *V* is the volume of the release medium taken at a specific interval time. *C_i_* and *C_n_* are the Pro concentrations (mg·mL^−1^) of *i* and *n* samples, respectively.

### 2.7. Stability Test

#### 2.7.1. Storage Stability of Pro@BMMs−PMAA/Fe^3+^ Nps

The storage stability of Pro@BMMs−PMAA/Fe^3+^ was tested according to the previous report [42,43]: 10 mg of Pro TC and 45.45 mg of Pro@BMMs−PMAA/Fe^3+^ were sealed in a 50 mL centrifuge tube and stored at 0, 25 ± 2, and 54 ± 2 °C for 14 days, respectively. The Pro contents were determined by HPLC and the changes in the concentration of Pro were calculated.

#### 2.7.2. Photostability of Pro@BMMs−PMAA/Fe^3+^ Nps

To determinate light stability of the samples under ultraviolet-light (UV-light) irradiation, approximately 5 mg of Pro@BMMs−PMAA/Fe^3+^ Nps was dispersed in 25 mL of PBS buffer containing 0.1% Tween-80 and exposed to a 36 W germicidal lamp (254 nm) at a distance of 20 cm in a UV-light incubator (WFH-203B, Shanghai Precision Instrument Co., Ltd., Shanghai, China). Samples were collected at given times and the Pro contents were analyzed by HPLC. The stability of Pro TC was studied using the same procedure.

### 2.8. Bioactivity Evaluation

In vitro bioactivity tests of Pro@BMMs−PMAA/Fe^3+^ Nps against *Rhizoctonia solani* were conducted using the growth-inhibition assay. Briefly, mycelium discs with a diameter of 5 mm were grown on the center of potato dextrose agar (PDA) plates, which were treated with different active ingredient contents (0, 0.0625, 0.125, 0.25, 0.5, and 1.0 mg·L^−1^) of Pro in the Nps. Pro TC solution and deionized water were used as control groups. The dishes were incubated at 25 °C and the colony diameters were measured separately for 10 days. The relative inhibition rates and the probit regression model of samples against *Rhizoctonia solani* were calculated according to the colony diameters.

For in vivo inhibitory efficacy tests, tomato Rhizoctonia rot was selected as the target with which to verify the preventive efficiency of Pro@BMMs−PMAA/Fe^3+^. Tomato leaves with good growth were selected for artificial inoculation of *Rhizoctonia solani*. The solutions of Pro@BMMs−PMAA/Fe^3+^ were diluted with 0.1% Tween-80 solution to a final Pro concentration of 1 mg·L^−1^ and sprayed onto fresh tomato leaves. After the evaporation of water on the tomato leaves, a 5 mm-diameter *Rhizoctonia solani* mycelial disc was inoculated on the tomato leaves and incubated at 25 °C. Pro TC and deionized water were used as controls. The diameters of lesions on the leaves were measured separately on the 5th and 10th days.

### 2.9. Uptake and Translocation of Nps

To study the delivery of Nps in vivo in fungi, 5 mm-diameter mycelium discs of Rhizoctonia solani were grown on PDA plates treated with 5 mg·mL^−1^ of BMMs−PMAA/5-AF. Deionized water was used as a control. The fungi were observed using a confocal laser microscope (CLSM, TCS-SP8 SR, Leica, Wetzlar, Germany) at 10 days.

To further explore the uptake and translocation of Nps in the plants, 5-AF-labeled Nps of BMMs−PMAA/5-AF were fabricated and utilized to handle the roots and leaves of tomato plants. To demonstrate the uptake and translocation of Nps through the roots, tomato plant seedlings were grown in 10 mL of nutrient solution with 50 mg of Nps at 30 °C in an artificial climate chamber (Kenton, Beijing, China). To verify that the Nps could be transferred to other tissues in the plant through leaves, 1 mL of BMMs−PMAA/5-AF suspension (5 mg·mL^−1^) was cautiously dripped and spread over one of the middle leaves. As a control, deionized water was employed to treat the plants in the test. A 12 h day–night alternation was conducted with a light intensity of 40 μW/cm^2^. Distributions of BMMs−PMAA/5-AF in various organizations (leaves, stems, and roots) of the tomato plants were observed using a fluorescence microscope (Axio Observer A1, ZEISS, Germany) after 24 h.

### 2.10. Nutritional Function of Pro@BMMs−PMAA/Fe^3+^ Nps

Rice seeds were sterilized and soaked in deionized water at 30 °C for 2 days to accelerate germination, and were cultivated at 30 °C in a dish (9 cm × 1.5 cm) with 50 mL of BMMs−PMAA/Fe^3+^ suspension (4 mg·mL^−1^) in the dark. Deionized water was employed as a blank control. After all the seeds germinated, the seedlings were grown in a chamber at 30 °C with 70% humidity in alternating light conditions, i.e., (12 h in day)/(12 h in dark). After the seedlings emerged, the effects of BMMs−PMAA/Fe^3+^ on the growth of plants were recorded, and the roots were cut and freeze-dried for EDS analysis.

### 2.11. Biosafety Evaluation

Human bronchial epithelial (16HBE) cells were cultured in 1640 media with 10% FBS and 1% penicillin/streptomycin at 37 °C for 24 h. Then, 16HBE cells were treated with different concentrations (0, 31.25, 62.5, 125, 250, and 500 μg·mL^−1^) of BMMs−PMAA/Fe^3+^ nanoparticles for 24 h, and the cell viability was assayed using a cell counting kit (CCK8, Dojindo, Japan). Deionized water was used as a control. Cell viability was calculated as follows: Cell viability (%) = (absorbance of treated group − absorbance of medium)/(absorbance of control group − absorbance of medium) × 100%.

*Escherichia coli* (*E. coli*) was cultured in Luria-Bertani (LB) culture media with different concentrations (0, 31.25, 62.5, 125, 250, and 500 μg·mL^−^^1^) of BMMs−PMAA/Fe^3+^ nanoparticles at 37 °C for 24 h, and the optical density (OD) values of *E. coli* were determined by UV-Vis absorptiometry (Eppendorf biophotometer plus, Hamburg, Germany) at 600 nm.

### 2.12. Data Analysis

The data were analyzed using statistical analysis software (SPSS 20.0, Chicago, IL, USA). All experiments were conducted three times. Statistical significance was measured by *p* < 0.05.

## 3. Results and Discussion

### 3.1. Preparation and Characterization of Pro@BMMs−PMAA/Fe^3+^ Nps

The BMMs was synthesized by the sol-gel method with CTAB and TEOS under basic conditions, and they were then modified with MPTES and methacrylic acid in turn to obtain BMMs−PMAA, which would afford ligands to the consequent coordination with Fe^3+^. Subsequently, after adding BMMs−PMAA to FeCl_3_ aqueous solution, “PMAA/Fe^3+^” with a metal cross-linked structure was deposited onto the surface of BMMs to form BMMs−PMAA/Fe^3+^. Finally, Pro was loaded into BMMs−PMAA/Fe^3+^ by impregnation adsorption, forming the coordination–bond response release formulation (Pro@BMMs−PMAA/Fe^3+^). The procedure is shown in Figure 1. The introduction of metal cross-linking structures could regulate pesticide release, and the utilization of Fe^3+^ as a cross-linking agent endowed the Nps with a stimulative effect on plant growth.

The morphologies of BMMs and Pro@BMMs−PMAA/Fe^3+^ were characterized by SEM. As shown in Figure 2A, the smooth spherical morphologies of Pro@BMMs−PMAA/Fe^3+^ Nps were similar to those of BMMs, indicating that modifying “PMAA/Fe^3+^” layers and the loading of Pro did not destroy the BMMs morphology. The average size of the BMMs was calculated to be approximately 20–50 nm by statistical analysis (Nano Measurer 1.2.5, Beijing, China), which was significantly lower than the hydrated particle size due to hydrate or aggregates of Nps in the solution [44]. We further tested the distributions of Fe and Cl in Pro@BMMs−PMAA/Fe^3+^ Nps by EDS analysis (Figure 2B). The presences of these elements demonstrated that Fe^3+^ and Pro were loaded in Pro@BMMs−PMAA/Fe^3+^ Nps.

As shown in Figure 3A, the size of Pro@BMMs−PMAA/Fe^3+^ was ~559 nm (PDI = 0.453), which was larger than that of BMMs, due to the grafting of “PMAA/Fe^3+^” layers and the loading of Pro. The N_2_ adsorption/desorption isotherms of Nps belong to Langmuir IV with two hysteresis loops (between 0.2–0.4 and 0.7–0.9 of P/P_0_), which confirmed the bimodal mesoporous structure of BMMs. As depicted in Figure 3B, with step-by-step modification and pesticide loading, the nitrogen adsorption capacity of each group of samples decreased successively, as did the corresponding specific surface area and pore volume. The surface area and pore volume of BMMs were the largest, 1161.66 m^2^·g^−1^ and 1.84 cm^3^·g^−1^, respectively. After surface modification, the specific surface area and pore volume of BMMs−PMAA/Fe^3+^ Nps dropped to 505.09 m^2^·g^−1^ and 0.75 cm^3^·g^−1^, respectively (Table 1). However, the shape of the hysteresis ring did not change much, indicating that the original mesoporous structure is still maintained. After Pro molecule loading, the surface area and pore volume of Pro@BMMs−PMAA/Fe^3+^ significantly decreased to 116.77 m^2^·g^−1^ and 0.47 cm^3^·g^−1^, respectively. More importantly, according to the BET pore size distribution model, the smaller mesoporous peaks of Pro@BMMs−PMAA/Fe^3+^ Nps disappeared. These findings revealed that the BMMs Nps were successfully covered by “PMAA/Fe^3+^” layers and the pores were occupied by Pro. In addition, the size of the simulated Pro molecule was approximately 1.03 nm (length), 0.98 nm (width), and 0.42 nm (height), as measured by ChemBio3D Ultra 12.0 (Suzhou C&J Marketing Co., Ltd, Suzhou, China) (Appendix A), showing that Pro could be conveniently loaded, stored, and released in BMMs−PMAA/Fe^3+^ Nps [45].

XPS analysis was performed to identify the chemical elements on the surface of the Nps (Figure 3C). In the BMMs spectrum, the binding energies of approximately 104.2 and 531.5 eV belong to Si 2p and O 1s, respectively. The weak signal at 283.3 eV corresponds to the C 1s originating from the residual carbon after calcination to eliminate the template of CTAB. Furthermore, the C 1s peak of BMMs−PMAA was more intense than that of BMMs, demonstrating the conjugation of PMAA onto the surface of BMMs. The typical characteristic peak of Fe^3+^ could be observed in the XPS spectrum of BMMs−PMAA/Fe^3+^ (Figure 3C,D), and the peaks between 712 and 724.8 eV were assigned to Fe^3+^, whereas other peaks were assigned as Fe^2+^ characteristic peaks and satellite peaks [46]. Compared to the spectrum of BMMs−PMAA/Fe^3+^, the two new signals in the XPS spectrum of Pro@BMMs−PMAA/Fe^3+^ with binding energies of 399.5 and 200 eV correspond to N 1s and Cl 2p, respectively, indicating that Pro was successfully loaded into BMMs−PMAA/Fe^3+^.

The structure of Pro@BMMs−PMAA/Fe^3+^ Nps was also tested by XRPD. As shown in Figure 4A, the (100) diffraction peak at 2θ = 1.94° was the characteristic peak of BMMs, indicating that it had a highly ordered mesoporous structure. After coating with “PMAA-Fe^3+^” layers, the characteristic diffraction peak intensity of BMMs−PMAA/Fe^3+^ was markedly decreased. The 2θ angle increased (1.90°–1.98°) and the corresponding *d* value decreased from 4.65 to 4.46 nm, indicating that the introduction of organics and metal ions had a certain influence on the mesoporous structure of BMMs. In addition, after BMMs−PMAA/Fe^3+^ Nps were loaded with Pro, the (100) diffraction peak of the spectrum disappeared, which meant that Pro molecules were successfully loaded into the mesoporous channel of BMMs.

Figure 4B shows that the zeta potential of BMMs was −29.1 mV owing to the de-protonation of Si-OH in neutral solution, and it then slightly increased to −28.2 mV after double-bond functional-group modification. However, after subsequent PMAA grafting, the zeta potential abruptly dropped to −39.7 mV. After iron chelation and loading of Pro, the zeta potential increased to −28.5 (BMMs−PMAA/Fe^3+^) and −28.0 mV (Pro@BMMs−PMAA/Fe^3+^), respectively, which was caused by the introduction of iron cations and the positive zeta potential of Pro [38]. FTIR measurement was performed to further investigate the functional groups and interactions existing in the nanosystem. As shown in Figure 4C, the absorption peak at 1075 cm^−1^ was attributed to the Si-O-Si stretching vibration. The peak of BMMs−MPS appearing at 1700 cm^−1^ was associated with C=O bonds. After grafting with PMAA, the spectrum of BMMs−PMAA exhibited a much stronger absorption band at approximately 1709 cm^−1^ due to more −COOH functional groups being introduced. Furthermore, Pro exhibits characteristic absorptions at 1558 and 1385 cm^−1^, which can be observed in Pro@BMMs−PMAA/Fe^3+^, indicating the successful loading of Pro into BMMs−PMAA/Fe^3+^. The TGA curves of all of the Nps are illustrated in Figure 4D. The weight loss in the range of 30–150 °C was attributed to the separation of water in the Nps, while that after 150 °C was due to the decomposition of the organic groups incorporated in the Nps. In the temperature range 150–800 °C, bared BMMs maintain a constant weight. However, the TGA curves of other samples showed a downward trend with increasing temperature. It could be clearly seen that after heating the samples up to 800 °C, BMMs−PMAA/Fe^3+^ and Pro@BMMs−PMAA/Fe^3+^ showed a gradually increased mass loss of 24.4% and 46.5%, respectively. Hence, the loading rate of Pro@BMMs−PMAA/Fe^3+^ was 22.1%, which was approximately consistent with the result of HPLC (24.0%). The TGA results were further evidence of the successful functionalization of BMMs with “PMAA/Fe^3+^” layers and loading of Pro into the Nps.

### 3.2. Release Behavior

The cumulative release behaviors of Pro@BMMs−PMAA/Fe^3+^ Nps were studied under different pH conditions. As shown in Figure 5, the release rate of Pro@BMMs−PMAA/Fe^3+^ at pH 7.0 after 144 h was 38.1%, indicating the good stability of Nps for preventing the premature release of Pro. Those at pH 5.0 and 9.0 were 62.3% and 84.1% at 144 h, respectively. These data suggested that Pro@BMMs−PMAA/Fe^3+^ Nps had better pH sensitivity, and the cumulative release rates of Pro under basic and weak acid conditions were higher than that under the neutral condition. Under the weak acid condition, the cross-linked framework of “PMAA/Fe^3+^” partially broke down due to the competitive bonding of −COO^−^ with Fe^3+^ and H^+^, which resulted in the “PMAA-Fe^3+^” coating becoming less tight [34]. Compared with the release under acidic conditions, the release rate of Pro@BMMs−PMAA/Fe^3+^ Nps under basic conditions was faster, and the overall cumulative release amount was greater. This was because the high concentration of hydroxide ions (OH^−^), which could weaken the complexation between −COO^−^ and metal ions, increased the dissociation degree of −COOH. The dissociated PMAA polymer chains repelled each other and moved away due to the increased electrostatic repulsion, resulting in the fast release of Pro under basic conditions [30,42].

To further elucidate the release mechanism, the release kinetics parameters of Pro@BMMs−PMAA/Fe^3+^ under pH values of 5.0, 7.0, and 9.0 were investigated with Zero-order, First-order, Higuchi, and Ritger–Peppas models. As shown in Table 2, the Ritger–Peppas model provided the most reasonable explanation of the Pro release mechanism, according to the correlation coefficient (*R*^2^) obtained. The fitted plots for Pro@BMMs−PMAA/Fe^3+^ using the Ritger–Peppas model were also illustrated in Figure 5B. The Ritger–Peppas model is based on a semi-empirical equation. When the release index (n) is less than 0.45, the mechanism mainly follows Fickian diffusion [33]. The calculated values of n at pH 5.0, 7.0, and 9.0 were 0.37, 0.31, and 0.35, respectively, indicating that the releases of Pro from Pro@BMMs−PMAA/Fe^3+^ nanoparticles in all kinds of situations were suitable for the Fickian diffusion. These results showed that pH had a significant effect on the release mechanism of Pro@BMMs−PMAA/Fe^3+^ Nps.

### 3.3. Stability Study

Stability is one of the important indicators for the pesticide evaluation system. The storage stability of Pro@BMMs−PMAA/Fe^3+^ was evaluated by measuring the active ingredient contents of Nps at different temperatures. Figure 6A shows that Pro@BMMs−PMAA/Fe^3+^ Nps remained stable during storage at 0, 25, and 54 °C for 14 days, and the levels of Pro in Nps were only decreased by 4.19% (0 °C) and 3.88% (54 °C), respectively, indicating that Pro in BMMs−PMAA/Fe^3+^ possessed good thermal stability.

One of the main reasons for the low utilization rate of many traditional pesticides is the degradation of pesticides by UV-light radiation. Therefore, it makes sense to encapsulate pesticides into sunscreen nanocarriers to avoid UV degradation. As shown in Figure 6B, Pro technical was extremely unstable under UV radiation, and the photodegradation rate was 94.6% after 24 h of UV irradiation. In contrast, Pro in Pro@BMMs−PMAA/Fe^3+^ Nps showed better photostability. Due to the protection of the BMMS−PMAA /Fe^3+^ carrier, the photodegradation rate of Pro in nanoparticles was only 39.7% after 24 h of UV radiation. With the extension of the radiation time, the degradation rate of Pro increased to 54.2% after 7 days. The improvement of photostability was mainly due to the fact that UV light could not reach Pro encapsulated in the channels of BMMs−PMAA/Fe^3+^ Nps, which could absorb or reflect UV light [47].

### 3.4. Bioactivity Evaluation

To determine if the Pro nano-delivery system influenced the biological activity in vitro, we examined the growth of *Rhizoctonia solani* in different Pro concentrations using the growth rate method. As shown in Figure 7A,B, compared with a blank control, the fungicidal activity of Pro@BMMs−PMAA/Fe^3+^ Nps was concentration-dependent and better than that of Pro technical. The inhibition rates were 90.3% (Pro@BMMs−PMAA/Fe^3+^ Nps) and 71.6% (Pro TC), respectively, at the Pro-as-an-active-ingredient concentration of 0.5 mg·L^−1^ at 10 days, due to the sustained release of Pro in the nano-delivery systems. Accordingly, the IC_50_ value of Pro@BMMs−PMAA/Fe^3+^ Nps (0.184 ± 0.013 mg·L^−^^1^) was lower than that of Pro technical (0.216 ± 0.080 mg·L^−1^). Many pathogenic fungi including *Rhizoctonia solani* can acidify the environment, and promote the release of pesticides in an acid-responsive nanosystem, which is consistent with the previous release result of Pro@BMMs−PMAA/Fe^3+^ Nps. In addition, the bioactivity of the BMMs−PMAA/Fe^3+^ carrier against *Rhizoctonia solani* was also determined, and the high concentration of BMMs−PMAA/Fe^3+^ Nps (3.6 mg·L^−1^) had no effect on the growth of fungi (Appendix A). These results indicated that Pro@BMMs−PMAA/Fe^3+^ Nps exhibited excellent fungicidal capability against *Rhizoctonia solani*.

The preventive efficacy of Pro@BMMs−PMAA/Fe^3+^ Nps against *Rhizoctonia solani* was further confirmed in tomato leaves (Figure 7C). When plants were challenged at 5 days post-spraying, both Pro@BMMs−PMAA/Fe^3+^ Nps and Pro technical treatments displayed an obvious pathogenicity reduction, compared with the negative control. The decay degree of the Pro@BMMs−PMAA/Fe^3+^ Nps treatment (2.17 mm back, 5 mm front) was slightly smaller than that of the Pro TC treatment (3.83 mm back, 6 mm front) (Figure 7D). When plants were cultivated for 10 days after inoculation of the fungi, the Pro@BMMs−PMAA/Fe^3+^ treatment still showed better effectiveness than Pro TC, and the rotten area of leaves obviously decreased, by 69.1% (back) and 33.3% (front), respectively. These results showed that the BMMs−PMAA/Fe^3+^ nanocarrier could reduce the photodegradation of loaded Pro molecules and facilitate intelligently targeted controlled release of the Pro active molecule.

### 3.5. Uptake and Translocation of BMMs−PMAA

To clearly explore the uptake and translocation of BMM-PMAA in fungi and tomato plant parts, 5-AF was grafted onto the BMMs−PMAA carrier surface. After a 10-day culture, the mycelia treated with BMMs−PMAA/5-AF at the edge of the colony exhibited obvious fluorescence signals under CLSM, as shown in Figure 8A. However, fluorescent signals were not detected in the blank sample. We further verified the uptake and translocation of Nps in the plants using a fluorescence microscope. Tomato plant seedlings were treated with BMMs−PMAA/5-AF in a hydroponic system. The leaves and roots of the tomato plants were treated separately with the BMMs−PMAA/5-AF suspension to ensure the translocation of nanocarriers through these two types of tissue to other parts. The BMMs−PMAA/5-AF suspension matched that used in the pesticide exposure test. The distribution of BMMs−PMAA/5-AF in roots, stems, and leaves of the tomato plants was observed by a fluorescence microscope. As shown in Figure 8B, no fluorescent signals were found in the blank control. However, the roots and stems in the plant root hydroponically treated with BMMs−PMAA/5-AF exhibited an obvious fluorescence signal, and the leaves revealed a weak fluorescence signal under the fluorescence microscope (Figure 8B-b). In addition, the fluorescent signals were also found in the roots and stems after the leaves were treated with Nps (Figure 8B-c). Unfortunately, no signals were found in the adjacent leaves. These results indicated that BMMs−PMAA Nps could not only be taken up by plant tissues, such as roots, leaves, etc., but could be translocated in these tissues as well. This might be due to the 20–50 nm size of BMMs, and the PMAA-modified BMMs Nps could serve as a carrier to deliver pesticides into plants and targeted microbes, which is similar to previous reports [23,24,25,26].

### 3.6. Nutritional Function of BMMs−PMAA/Fe^3+^

Nano-pesticides possess multiple functions, such as fungicidal action and nutrient supplementation, which are of great significance in agricultural applications [17,48]. To further explore the effect of BMMs−PMAA/Fe^3+^ used as an iron fertilizer for enhancing plant growth, pot experiments were conducted. As shown in Figure 9, no significant differences were found in the germination of rice seeds treated with BMMs−PMAA/Fe^3+^ and the blank control on the second day. The heights of the seedlings treated with BMMs−PMAA/Fe^3+^ were 4.5% and 10.8% higher than that of water at 16 and 23 days, respectively. Furthermore, the leaves of rice seedlings treated with nanocarriers were larger and healthier, and their root systems were more developed. EDS analysis of the rice roots was further carried out to determine the content of elemental C, O, N, Si, and Fe (Appendix A). EDS scanning results showed that the Fe content in the roots of rice cultivated with BMMs−PMAA/Fe^3+^ Nps was much higher than that of the blank control. These findings clearly indicated that BMMs−PMAA/Fe^3+^ Nps could provide crop trace metal elements for the growth and development of rice. In addition, it also proved the safety of using BMMs−PMAA/Fe^3+^ as a pesticide carrier for crop growth.

### 3.7. Biosafety Evaluation

We performed the biosafety evaluations of BMMs−PMAA/Fe^3+^ nanocarriers using toxicological methods. Figure 10 shows that BMMS-PMAA/Fe^3+^ Nps with different concentrations displayed no obvious influence on the growth of *E. coli* and 16HBE cells. On the contrary, the bioactivities of *E. coli* and cells were promoted due to the introduction of an iron trace element [49,50]. As a result, BMMs−PMAA/Fe^3+^ nanocarriers possessed desirable biocompatibility and high biosafety.

## 4. Conclusions

In this study, we constructed a pesticide-delivery system based on BMMs coated with the “PMAA/Fe^3+^” gatekeeper. The obtained nano-pesticide Pro@BMMs−PMAA/Fe^3+^ had a high pesticide-loading rate of 24.0% and exhibited excellent UV-shielding and thermal performance. The release of Pro from BMMs−PMAA/Fe^3+^ was pH dual-sensitive due to the existence of metal coordination between −COO^−^ and Fe^3+^. The cumulative release rates under basic conditions (pH 9.0) and weak acid conditions (pH 5.0) were higher than that under neutral conditions (pH 7.0). The Ritger–Peppas kinetic model was able to describe the sustained release curves. Biological activity experiments showed that Pro@BMMs−PMAA/Fe^3+^ nanoparticles (Nps) have excellent antifungal activity against *Rhizoctonia solani* compared to Pro TC at the same concentration of active ingredient, which provided a novel defense measure to crops against wilt disease. The uptake and translocation of nanocarriers in tomato plants and fungi mycelia were confirmed by fluorescence tracking of 5-AF-labeled Nps, demonstrating that BMMs−PMAA can serve as a vehicle to deliver pesticide molecules into plants and mycelia. Moreover, Pro@BMMs−PMAA/Fe^3+^ Nps had excellent biological safety and promoted the growth of crops as an iron micronutrient. Taken together, these results can provide design strategies for obtaining an environmentally friendly multifunctional nano-pesticide delivery system and promote the development of green agriculture in the future.

## Figures and Tables

**Figure 1 nanomaterials-12-00687-f001:**
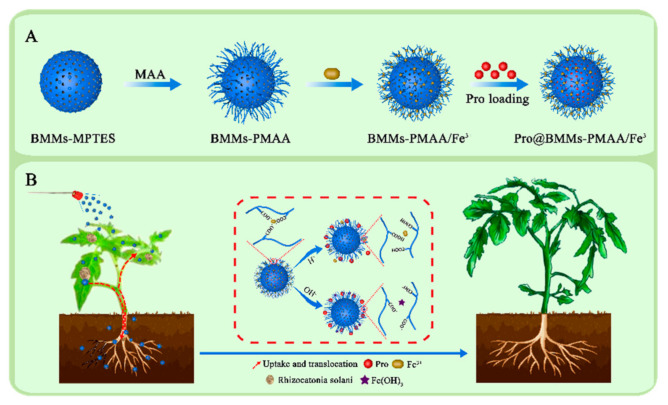
Schematic illustration of the preparation of Pro@BMMs−PMAA/Fe^3+^ nanoparticles (**A**) and the applications in improving plant nutrition and protection (**B**).

**Figure 2 nanomaterials-12-00687-f002:**
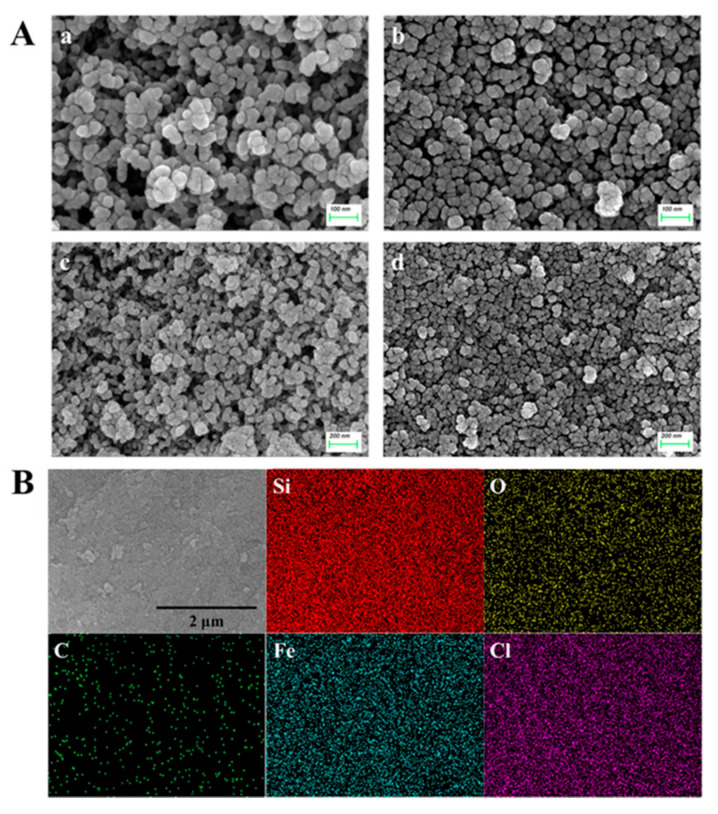
SEM images of the BMMs (**A**-a,**A**-c) and Pro@BMMs−PMAA/Fe^3+^ (**A**-b,**A**-d) and EDS mapping characterizations (**B**) of Pro@BMMs−PMAA/Fe^3+^ nanoparticles.

**Figure 3 nanomaterials-12-00687-f003:**
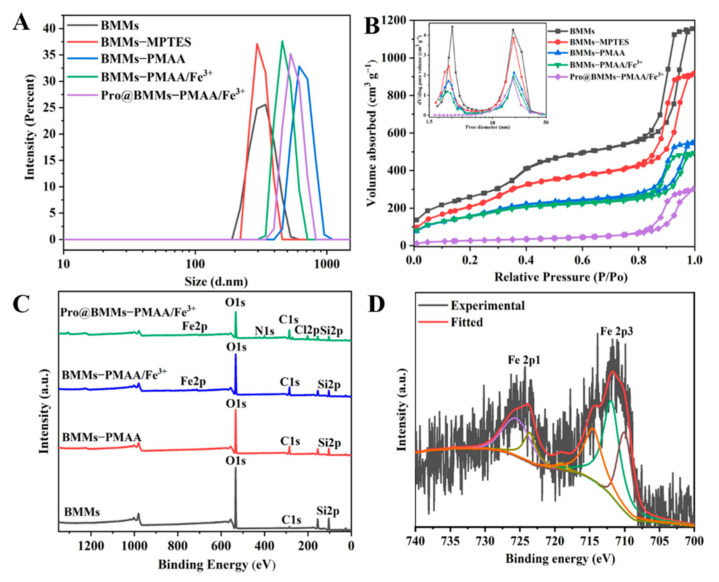
Size distributions (**A**), N_2_ adsorption/desorption isotherms (**B**), XPS survey spectra (**C**), and curve fittings of Fe 2p peak (**D**) of Pro@BMMs−PMAA/Fe^3+^ Nps and control samples.

**Figure 4 nanomaterials-12-00687-f004:**
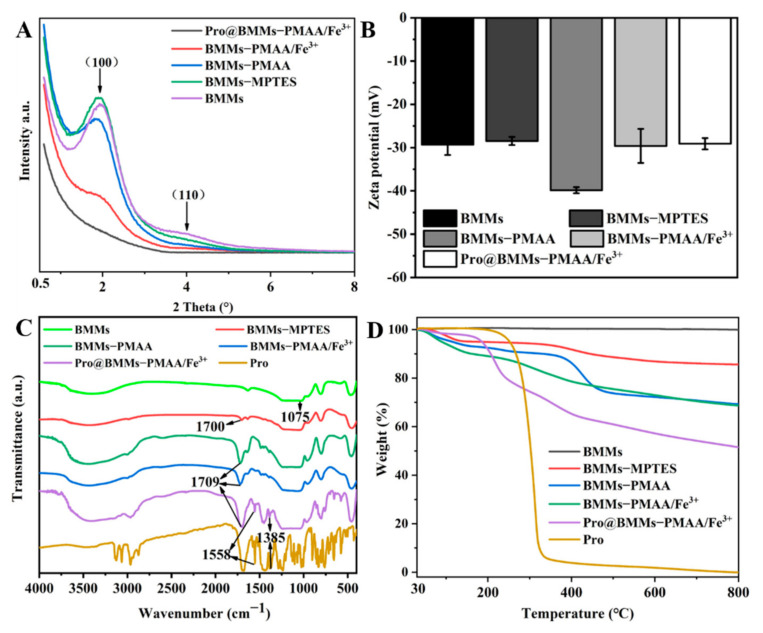
XRPD diffractograms (**A**), zeta potentials (**B**), FTIR spectra (**C**), and TGA curves (**D**) of BMMs, BMMs−MPTES, BMMs−PMAA, BMMs−PMAA/Fe^3+^, and Pro@BMMs−PMAA/Fe^3+^ Nps.

**Figure 5 nanomaterials-12-00687-f005:**
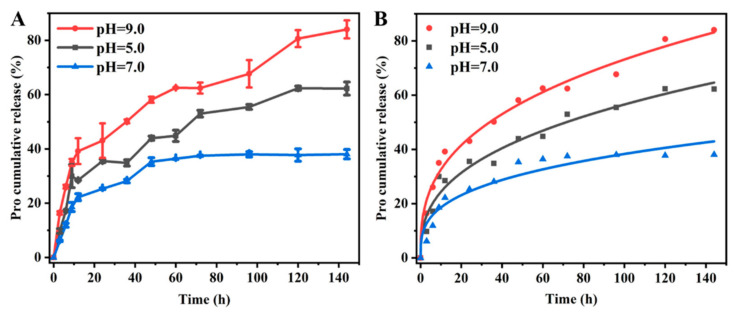
Cumulative release behaviors (**A**) and fitting plots using the Ritger–Peppas equation (**B**) of Pro@BMMs−PMAA/Fe^3+^ Nps at different pH values.

**Figure 6 nanomaterials-12-00687-f006:**
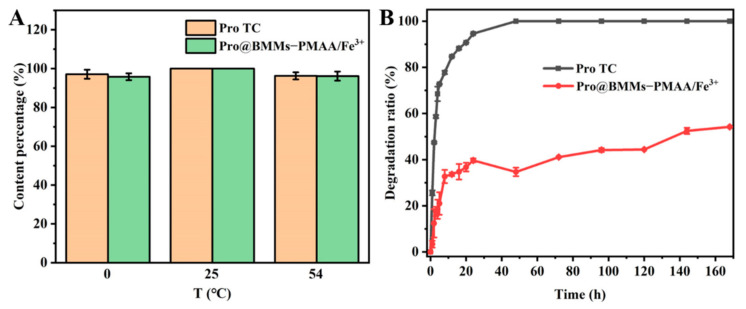
The storage stability (**A**) and the photostability (**B**) of Pro@BMMs−PMAA/Fe^3+^ Nps.

**Figure 7 nanomaterials-12-00687-f007:**
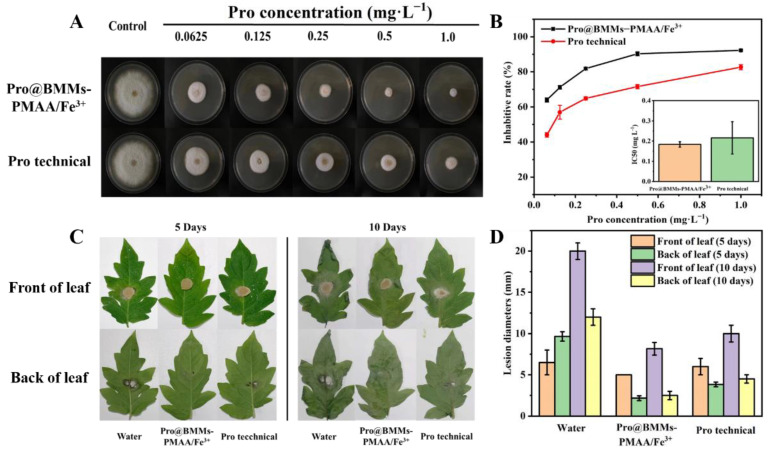
Bioactivity evaluation of Pro@BMMs−PMAA/Fe^3+^ nanoparticles. Digital images (**A**), and inhibitory rates and IC_50_ values (**B**) of Pro@BMMs−PMAA/Fe^3+^ Nps in vitro against *Rhizoctonia solani*. The effects of Pro@BMMs−PMAA/Fe^3+^ nanoparticles on the lesion development by *Rhizoctonia solani* on tomato leaves (**C**,**D**). Lesion diameter was measured at 5 and 10 days after the fungi inoculation.

**Figure 8 nanomaterials-12-00687-f008:**
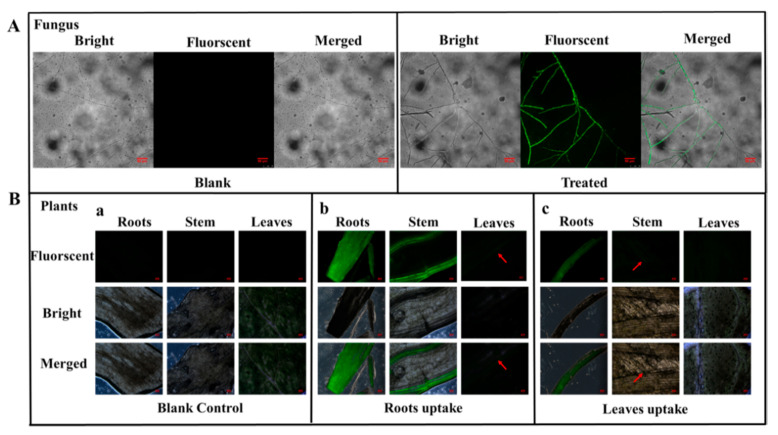
Confocal imaging of *Rhizoctonia solani* incubated with BMMs−PMA/5-AF for 10 days (**A**). Fluorescence microscope images of different parts of the tomato plant (leaves, stems, and roots) after 1 day of treatment with BMMs−PMAA/5-AF for the roots (**B-b**) and the leaves (**B-c**) compared to a blank control (**B-a**).

**Figure 9 nanomaterials-12-00687-f009:**
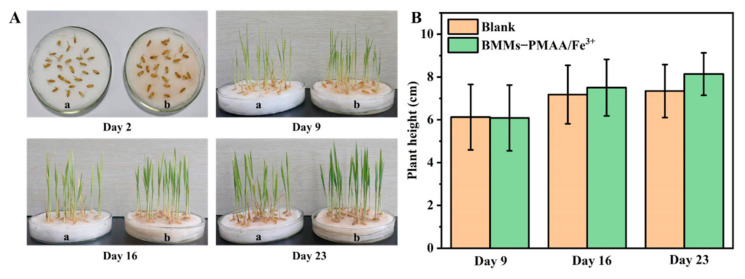
Growth photographs (**A**) and heights (**B**) of rice plant with BMMs−PMAA/Fe^3+^ treatment on different days.

**Figure 10 nanomaterials-12-00687-f010:**
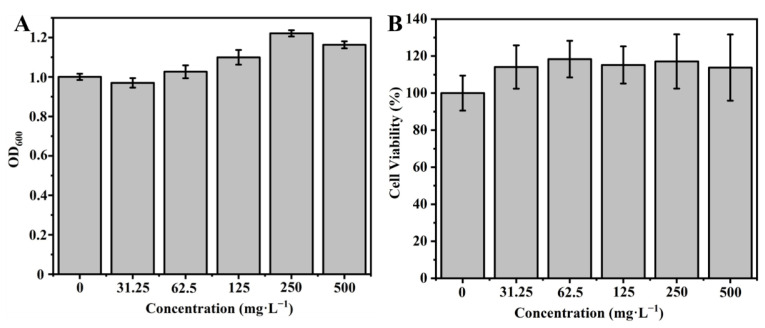
Biosafety evaluations of different concentrations of BMMs−PMAA/Fe^3+^. The OD_600_ values of *E. coli* suspension (**A**), and biological activities of 16HBE cells (**B**).

**Table 1 nanomaterials-12-00687-t001:** BET and BJH measurements of the samples.

Sample	S_BET_ (m^2^·g^−1^)	Pore Volume (cm^3^·g^−1^)	Small Pore (nm)	Large Pore (nm)
BMMs	1161.66	1.84	2.99	18.52
BMMs−MPTES	978.04	1.47	2.69	18.81
BMMs−PMAA	622.73	0.91	2.67	18.73
BMMs−PMAA/Fe^3+^	505.09	0.75	2.57	18.69
Pro@BMMs−PMAA/Fe^3+^	116.77	0.47	-	18.42

**Table 2 nanomaterials-12-00687-t002:** Release kinetics parameters of Pro@BMMs−PMAA/Fe^3+^ Nps at different pH values.

Fitting Models	pH Values	Kinetic Equations	*R* ^2^
Zero-order	5.0	Q = 0.37t + 18.65	0.80
	7.0	Q = 0.23t + 14.79	0.63
	9.0	Q = 0.47t + 25.10	0.82
First-order	5.0	Q = 1–1.22 e^0.006t^	0.89
	7.0	Q = 1–1.27 e^0.011t^	0.95
	9.0	Q = 1–1.18 e^0.003t^	0.68
Higuchi	5.0	Q = 5.10 t^1/2^ + 6.32	0.95
	7.0	Q = 3.29 t^1/2^ + 6.16	0.86
	9.0	Q = 6.47 t^1/2^ + 9.52	0.96
Ritger–Peppas	5.0	Q = 10.47 t^0.37^	0.97
	7.0	Q = 9.02 t^0.31^	0.92
	9.0	Q = 14.49 t^0.35^	0.99

Note: Q is the fractional release of pesticide, t is the elapsed time, e is natural logarithm and *R*^2^ is the high value of the linear regression coefficient.

## Data Availability

Not applicable.

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
