# Peer review of "A pH Dual-Responsive Multifunctional Nanoparticle Based on Mesoporous Silica with Metal-Polymethacrylic Acid Gatekeeper for Improving Plant Protection and Nutrition"

_nanomaterials, 2022, doi:10.3390/nano12040687_

Round 1

Reviewer 1 Report

In this work authors explored an acid/base duel-responsive nanosystem using BMMs-PMAA/Fe3+ as carriers to deliver prochloraz for control plant protection and environmental safety. The work is worthy of attention and interesting to the reader. It contains original results and the experimental work has been carefully designed and carried out. Results are presented in a clear way  and exhaustively support discussion. Conclusion is congruent with the experimental work and results.

Author Response

Thanks for you comments.

Reviewer 2 Report

The manuscript titled “A pH dual-responsive multifunctional nanoparticle based on mesoporous silica with metal-polymethacrylic acid gatekeeper for improving plant protection and nutrition is reviewed for consideration in nanomaterials. The research methodology is sound, and results looks amazing. However,

In section 2.7 Photostability behavior, the HPLC methodology is not clear, please mention all parameters.

Comment: The conclusion part is not up to point, please be specific and to the point.

Reviewer 3 Report

The manuscript proposes the use of silica nanoparticles as carriers of prochloraz funcicide for slow release over time. Several steps of nanoparticle finctionalization are herein described and the reader is easily confused. The authors must reorganized their works and define the term ‘control’ because many times water is considered as control and in  toxicological studies several series of controls are implemented. For example, for UV VIS experiments, 24h is not sufficient and systematic investigations must be carried out after 1 week, one month, etc.

Additionally, after reading the reported data, readers are concerned about the future toxicity of rice treated with BMMS-modified NPs. Reported toxicology studies using tomato leaves and fungi are not relevant because rice plants treated with silanized modified nanoparticles may induce unexpected side effects in humans over time. The main question therefore is whether pure insecticide is the best choice than nanoparticles modified with pesticide, as very few studies are available to assess potential disease in humans due to silica NP accumulation.

Other remarks:

  • Information about BMMS is not provided in materials and methods
  • As above, the main reviewer’s main concern relates to the toxicity of the organic preparation of BMM_MPTES and, secondly, their additional chemical formulation with Prochloraz (Pro), a broad-spectrum, high-efficiency imidazole fungicide. Any possible resistance in the presence of antibiotics? What is the stability of modified BMM in real time?
  • Line 209: what does 10d mean?
  • Figure 2A and B need to be explained. SEM images are colored and there are not clear. Please provide better quality SEM images of different sizes (e.g. 5 um x5 um). AFM images are also encouraged. Also mention if the images are recorded from different areas.

Round 2

Reviewer 3 Report

The authors responded reasonably to the reviewer's comments.